# Domain-independent Plan Intervention When Users Unwittingly Facilitate Attacks

**Sachini Weerawardhana** and **Darrell Whitley**[1] **Mark Roberts**[2]

[1]Computer Science Department, Colorado State University, Fort Collins, CO, USA | {sachini, whitley}@cs.colostate.edu
[2]The U.S. Naval Research Laboratory, Code 5514; Washington, DC, USA | mark.roberts@nrl.navy.mil

## Abstract

In competitive situations, agents may take actions to achieve their goals that unwittingly facilitate an opponent's goals. We consider a domain where three agents operate: (1) a user (human), (2) an attacker (human or a software) agent and (3) an observer (a software) agent. The user and the attacker compete to achieve different goals. When there is a disparity in the domain knowledge the user and the attacker possess, the attacker may use the user's unfamiliarity with the domain to its advantage and further its own goal. In this situation, the observer, whose goal is to support the user may need to intervene, and this intervention needs to occur online, on-time and be accurate. We formalize the online plan intervention problem and propose a solution that uses a decision tree classifier to identify intervention points in situations where agents unwittingly facilitate an opponent's goal. We trained a classifier using domain-independent features extracted from the observer's decision space to evaluate the "criticality" of the current state. The trained model is then used in an online setting on IPC benchmarks to identify observations that warrant intervention. Our contributions lay a foundation for further work in the area of deciding when to intervene.

## 1   Introduction

When an agent is executing a plan to achieve some goal, it's progress may be challenged by unforeseen changes such as an unexpected modification to the environment or an adversary subverting the agent's goal. In these situations, a passive observer intervening to help the agent reach it's intended goal will be beneficial. Intervention is different from the typical plan recognition problem because we assume the observed agent pursues desirable goals while avoiding undesirable states. Therefore, the observer must (1) monitor actions/state unobtrusively to predict trajectories of the observed agent (keyhole recognition) and (2) assist the observed agent to safely complete the intended task or block the current step if unsafe. Consider a user checking email on a computer. An attacker who wants to steal the user's password makes several approaches: sending an email with a link to a phishing website and sending a PDF file attachment embedded with a keylogger. The user, despite being unaware of the attacker's plan, would like to complete the task of checking email safely and avoid the attacker's goal. Through learning, our observer can recognize risky actions the user may execute in the environment and ensure safety.

The decision of when to intervene must be made judicially. Intervening too early may lead to wasted effort chasing down false positives, helpful warnings being ignored as a nuisances, or leaking information for the next attack. Intervening too late may result in the undesirable state. Further, we are interested in assisting a human user with different skill levels, who would benefit more from customized intervention. To this end, we need to identify actions that warrant intervention over three different time horizons: (1) critical action, which if unchecked will definitely trigger the undesirable state, (2) mitigating action, which gives the user some time to react because the threat is not imminent and (3) preventing actions, which allows for long term planning to help the user avoid threats. Based on the time horizon we are current in, we can then plan to correct course accordingly. In this work we focus on identifying the first horizon. Intervention is useful in both online settings, where undesirable states may arrive incrementally and in offline settings where observations are available prior to intervention.

In this paper, we model online intervention in a competitive environment where three agents operate: (1) a user (human), (2) an attacker (human or a software) agent and (3) an observer (a software) agent who will intervene the user. The observer passively monitors the user and the attacker competing to achieve different goals. The attacker attempts (both actively and passively) to leverage the progress made by a user to achieve its own goal. The attacker may mask domain knowledge available to the user to expand the attack vector and increase the likelihood of a successfull attack. The user is pursuing a desirable goal while avoiding undesirable states. Using domain-independant features, we train a decision tree classifier to help the observer decide whether to intervene. A variation of the relaxed plan graph (Blum and Furst 1997) models the desirable, undesirable and neutral states that are reachable at different depths. From the graph, we extract several domain independent features: risk, desirability, distances remaining to desirable goal and undesirable states and active landmarks percentage.

We train a classifier to recognize an observation as a intervention point and evaluate the learned model on previously unseen observation traces to assess the accuracy. Furthermore, the domain independent features used in the classifier offer a mechanism to explain why the intervention occurred. In real-time, making the decision to intervene for each ob-

servation may be costly. We examine how the observer can establish a wait time without compromising accuracy.

The contributions of this paper include: (1) formalizing the online intervention problem as an intervention graph that extends the planning graph, (2) introducing domain-independent features that estimate the criticality of the current state to cause a known undesirable state, (3) presenting an approach that learns to classify an observation as intervention or not, (4) incorporating salient features that are better predictors of intervention to generate explanations, and (5) showing this approach works well with benchmarks.

## 2    Example

Before we formalize the problem, we present examples for two cases of the online intervention: (1) the attacker is actively trying to make the user reach the undesirable state by leveraging the user's progress and (2) the passive attacker introduces an undesirable state to the environment without the user's knowledge (i.e., a trap), where attacker masks the location of the trap and exploits the user's unfamiliarity with the domain to make the user reach the undesirable state. In both cases, the observer monitors the attacker and the users' actions. The user plans for a desirable goal state, $G_d$. Given the unexpected modification to the domain model, executing this plan may likely cause the user to reach the undesirable state ($G_u$). The observer is assumed to be familiar with the domain (regardless of attacker's attempts to mask information to the user) and has knowledge about commonly occurring goals such as $G_d$ and $G_u$. The user would like to be interrupted if some action will trigger $G_u$.

**Active Attacker**: We use the IPC block-words domain (Gupta and Nau 1992) to illustrate the active attacker's case. The observer is watching the user stacking blocks to spell a word. The domain contains 4 blocks: T, B, A, D. Figure 1 shows the undesirable state developing from initial state $I$. $G_d$ equals the word TAD, while $G_u$ equals the word BAD. The user can not recognize block B (indicated by dotted lines), which prevents the user from identifying states resulting from performing operations on B such as stack and pick up, and therefore fail to circumvent $G_u$ on his own. The attacker will use block B to defeat the user and achieve $G_u$.

In the initial state ($I$), all blocks are on the table. The user's arm (solid line) and the attacker's arm (dotted line) are empty. In the next sequence of events, the observer sees that the user has picked up block A ($S_1$) and stacked A on D ($S_2$). Consider two alternative timelines $T_1$ and $T_2$ stemming from $S_2$. In $T_1$, the observer sees that the user has picked up T and the attacker has also picked up B. The next state shows that the user has stacked T on A to spell the word TAD and reached $G_d$ successfully. In timeline $T_2$, the attacker has succeeded in reaching $G_u$ by stacking B on A before the user stacked T on A, leveraging the user's progress.

**Passive Attacker**: This case considers the 3x3 grid world domain (McDermott 1999) shown in Figure 2. The observer watches the user (white circle) navigating from a start point (0,0) on the grid to reach $G_d$ point (3,3) in 1-step actions. When executing a plan to reach $G_d$, the user would like to avoid the trap at point X (2,3), $G_u$ but will not be able to

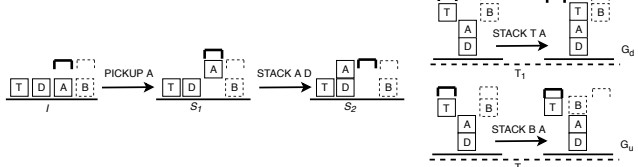

Figure 1: Reaching $G_u$ with an active attacker

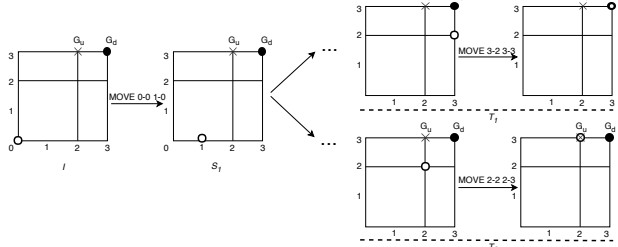

Figure 2: Reaching $G_u$ with passive attacker

do so unless the observer interrupted. Let us assume the observer sees the user's action resulting in state $S_1$. Although the move indicates that the user is moving toward $G_u$ and $G_d$, interruption is too early. In two alternative timelines $T_1$ (top right) and $T_2$ (bottom), the observer sees different moves. In $T_1$ the user has reached $G_d$ while avoiding $G_u$, in which case the observer need not interrupt. However, in $T_2$ the user has reached $G_u$, in which case it would have been helpful if the user was blocked before moving to (2,3).

## 3    The Intervention Problem

Our formulation of the intervention problem makes several assumptions about the three actors. (1) **Observer**: intervention decisions are made in an online setting for each observation that appears incrementally and include actions executed by the attacker or the user. The goals $G_d$ or $G_u$ are known but the plans to reach $G_d$ or $G_u$ are hidden. The domains for which plan intervention problem is defined are discrete and all actions are assumed to be of unit cost. The observer has full observability in the domain and the environment is deterministic. Therefore, it can determine the actions that are immediately applicable in the current state. (2) **User**: Follows a plan to reach $G_d$, but may reach $G_u$ unwittingly. $G_u$ is hidden, but would like the observer's help to avoid $G_u$. The user does not have full observability of the domain or the attacker's actions. (3) **Attacker**: Follows a plan to reach $G_u$. The attacker has full observability of the domain and the user's actions. Given these assumptions, the observer assesses the state after each observation. This requires the observer to hypothesize about possible interesting trajectories from current state and evaluate each trajectory in terms of their likelihood to cause $G_u$.

### 3.1    Definitions

Following STRIPS (Fikes and Nilsson 1971), we define a planning problem as a tuple $P = \langle F, A, I, G \rangle$ where $F$ is the

set of fluents, $I \subseteq F$ is the initial state, $G \subseteq F$ represents the set of goal states and $A$ is the set of actions. Each action $a \in A$ is a triple $a = \langle Pre(a), Add(a), Del(a) \rangle$ that consists of preconditions, add and delete effects respectively, where $Pre(a), Add(a), Del(a)$ are all subsets of $F$. An action $a$ is applicable in a state $s$ if preconditions of $a$ are true in $s$; $pre(a) \in s$. If an action $a$ is executed in state $s$, it results in a new state $s' = (s \setminus del(a) \cup add(a))$. The solution to $P$ is a plan $\pi = \{a_1, \ldots, a_k\}$ of length $k$ that modifies $I$ into $G$ by execution of actions $a_1, \ldots, a_k$.

The plan recognition problem defined by Ramirez and Geffner (2010) is a triple $T = \langle D, \mathcal{G}, O \rangle$ where $D = \langle F, A, I \rangle$ is a planning domain, $\mathcal{G}$ is the set of goals, and $\mathcal{G} \subseteq F$. An observation sequence $O = o_1, \ldots, o_m$ are actions $o_i \in A, i \in [1, m]$. A solution to the plan recognition problem is a subset of goals $G \in \mathcal{G}$ for which an optimal plan $P[G]$ satisfying $O$ is produced.

Similarly, the plan intervention problem ($\mathcal{I}$) also uses observations of actions. However, instead of using information gleaned from the observation trace to find the most likely plans (and goals), the intervention problem aims to assess the current state for it's ability to cause $G_u$ and identify whether or not the user needs to be blocked from making further progress. Unlike Ramirez and Geffeners' approach, the observations used in our solution are not noisy nor do they contain missing actions. This will be addressed in future work.

**Plan intervention problem** $\mathcal{I} = \langle D, O, G_u, G_d, M \rangle$ consists of a planning domain $D$ and a sequence of observed actions $O$, a set of undesirable states $G_u \subseteq F$, a set of desirable states $G_d \subseteq F$ ($G_u \neq G_d$), and a decision tree classifier model $M$ that combines a vector of domain-independant features to classify an obervation as requiring intervention or not. The extension to typical plan/goal recognition comes from the domain-independent feature vector, which will be discussed in section 3.3. A solution to $\mathcal{I}$ is a vector of decision points corresponding to actions in $O$ indicating whether each action was identified as requiring intervention.

### 3.2 Modelling the Intervention Decision Space

To assess the criticality of the current state to cause $G_u$, the observer enumerates action sequences that will transform the current state to $G_d$. These action sequences and intermediate states make up the observer's decision space, which is a single-root directed acyclic connected graph $S = \langle V, E \rangle$, where $V$ is the set of vertices denoting possible states the user could be in until $G_d$ is reached, and $E$ is the set of edges representing actions from $A$. We refer to this graph as the *intervention graph*. The root of the intervention graph indicates the current state. Leaves of the graph are goal states (i.e., $G_u$ and $G_d$). A path from root of the tree to $G_u$ represents a candidate attack plan, while a path from root to leaf node containing $G_d$ represents a desirable plan.

Figure 3 illustrates the observer's decision space for unobserved actions extending from state $S_1$ in Figure 1. Some subtrees are hidden for simplicity. Given the initial state where all 4 blocks are on the table, the observer expects the next action to be one in the set (PICK-UP {T, D, A, B}), but B is hidden from the user. The attacker can execute any

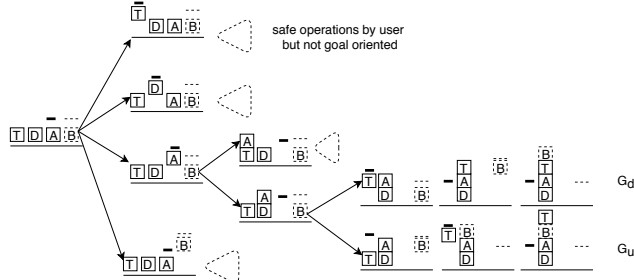

Figure 3: Fragment of the decision space at state $I$ for block-words plan intervention example in Figure 1

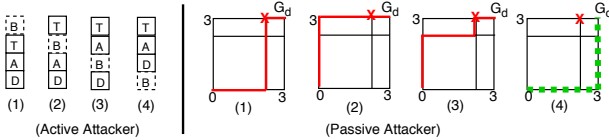

Figure 4: User achieving $G_d$ amid attacker actions in intervention examples in Section 2

of the 4 actions. Using the intervention graph, the observer hypothesizes all possible action sequences that can be observed in the future, that will lead to $G_u$ (spell BAD) or $G_d$ (spell TAD). One such sequence (as shown in the figure) is: PICK-UP A $\rightarrow$ STACK A D $\rightarrow$ PICK-UP T $\rightarrow$ STACK T A. At this point the user reaches $G_d$. On the other hand if the sequence was PICK-UP A $\rightarrow$ STACK A D $\rightarrow$ PICK-UP B $\rightarrow$ STACK B A, with the last two actions executed by the attacker, the attacker achieves $G_u$.

Figure 4 illustrates how the user's plans to reach $G_d$ could fail in the presence of an active (left) or passive (right) attacker. In the case of an active attacker, given the assumption that the attacker does not backtrack to a previous state and only leverages progress made thus far, it can make four attempts to prevent the user from reaching $G_u$ by inserting the hidden block into the partially built stack. If the user achieves goal states 1 or 4 the user wins despite the attacker. If the observed actions indicate that the user is heading toward one of these two states, then an interrupt is unwarranted. State 3 is less ideal for the user but $G_u$ is not achieved. In state 2 the attacker has successfully reached $G_u$. Observations leading to state 2 warrant interruption.

In the case of a passive attacker, the observer needs to hypothesize about likely goals of the user given the current state. Figure 4 (right) shows three of many such plans the user may follow to reach $G_d$. Paths 1, 2 and 3 all result in user going past the undesirable state (marked x), and at some point in these observation sequences the user must be interrupted before $G_u$ is reached. In contrast, path 4 indicates a safe path and must not generate an interrupt.

Algorithm 1 describes how the intervention graph is built. The intervention graph is similar to the relaxed planning graph (RPG), where each level consists of predicates that have been made true and actions $a \in A$ whose preconditions are satisfied. Initially, before any observations have

been made, the current state (i.e., root of the tree) is set to initial state $I$. Next, using the domain theory $D$, actions $a \in A$ whose preconditions are satisfied at current state are added to the graph. Each action in level $i$ spawn possible states for level $i+1$. Calling the method recursively for each state until $G_d$ and $G_u$ are added to some subsequent level in the graph will generate a possible hypotheses space for the observer. As a new observation arrives, the root of the graph is changed to reflect the new state after the observation and subsequent layers are also modified to that effect. Similar to the RPG, we omit delete effects during construction. Also construction terminates once $G_d$ is reached. The graph building algorithm does not allow adding backtracking actions because it will create a cycle.

---

**Algorithm 1** Build Intervention Graph

---

**Require:** $D$, $s$, $G_u$, $G_d$
1: $i = 0$; $s_i \leftarrow I$
2: **procedure** EXPANDGRAPH($D$, $s$, $G_u$, $G_d$)
3:     **if** $s_i \models G_u, G_d$ **then** return $\langle V, E \rangle$
4:     **else**
5:         **for** $a \in A$ where $Pre(a) \in s_i$ **do**
6:             $s_{i+1} \leftarrow ((s_i \setminus Del(a)) \cup Add(a))$
7:             **if** $s_{i+1} \equiv s_i$ **then** continue
8:             $v \leftarrow$ AddVertex ($s_{i+1}$)
9:             $e \leftarrow$ AddEdge ($s$, $s_{i+1}$, $a$)
10:             $V \cup \{v\}$ ; $E \cup \{e\}$
11:             ExpandGraph ($D$, $s_{i+1}$, $G_u$, $G_d$)

---

### 3.3 Domain Independent Features

We extract a set of features from the intervention graph that help determine when to intervene. These features include: Risk, Desirability, Distance to $G_d$, Distance to $G_u$ and Percentage of active undesirable landmarks in current state. We use these features to train a decision tree. Figure 5 illustrates a fragment of the intervention graph after PICK-UP A. Following the subtree extending from action STACK A D, both $G_u$ and $G_d$ can be reached. Unexpanded subtree $T_1$ also contains instances where the user can reach $G_d$ safely, without reaching $G_u$. We will use Figure 5 as a running example to discuss feature computation.

**Risk** ($R$) quantifies how likely the effects of current observation will lead to $G_u$. $R$ is also coupled with the uncertainty the observer has about the next observation. We model the uncertainty as a uniform probability distribution across the set of actions whose preconditions are satisfied in current state. We define $R$ as the posterior probability of reaching $G_u$ while the user is trying to achieve $G_d$. Given the intervention graph, we extract paths from root to any leaf containing the $G_d$, including the ones in which the user has been subverted to reach $G_u$ instead. By virtue of construction termination, $G_d$ will always be a leaf. $R$ is computed for paths leading to state (2) in Figure 4 (left) because in that state the attacker has won. In the passive attacker case any path in the intervention graph that causes the user to reach point X, before $G_d$ is reached qualifies as candidates to compute $R$.

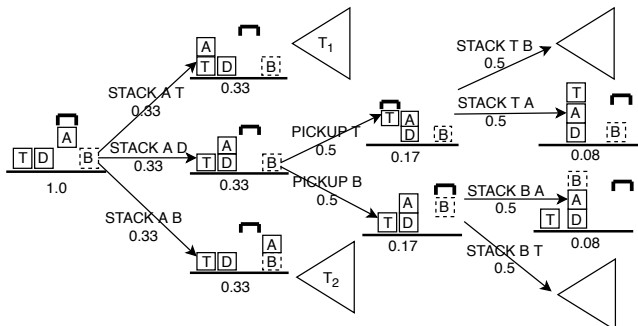

Figure 5: Fragment of the decision space after PICKUP A has been observed for block-words example. Numbers under each state and action indicate the probability. Subrees $T_1$ and $T_2$ are not expanded for simplicity.

Let $\Pi_{candidates}$ be the plans reaching $G_d$ and let $|\Pi_{candidates}| = n$. The plan set $\Pi_u$ contains action sequences that reach state $G_u$ such that, $\Pi_u \subseteq \Pi_{candidates}$, $|\Pi_u| = m$ and ($m <= n$). We compute posterior probability of reaching $G_u$ for a path $\pi \in \Pi_u$, using chain rule in probability as, $P_\pi = \prod_{j=1}^{k} P(\alpha_j | \alpha_1, \alpha_2, ..., \alpha_{k-1})$, and $\alpha_j \in A$ and $k$ is the length of path until $G_u$ is reached. Then:

$$R = \begin{cases} \frac{\sum_{i=1}^{m} P_{\pi_i}}{m} & m > 0 \\ 0 & m = 0 \end{cases}$$

There are six action sequences the observer might observe when the user is trying to achieve $G_d$ ($n = 6$) and only one of those six sequences will make the user reach $G_u$ ($m = 1$). Since we assumed full observability for the observer, the root of the tree (current state) is assigned the probability of 1.0. Then, actions that are immediately possible after current state (STACK A B, STACK A D, STACK A T) are each assigned probabilites following a uniform distribution across the branching factor (0.33). Then for each applicable action in the current state, the resulting state gets the probability of ($1.0 \times 0.33 = 0.33$). Similarly, we apply the chain rule of probability for each following state and action level in the graph until $G_u$ first appears in the path. In this graph, $G_u$ appears two actions later and $R = \frac{0.08}{1} = 0.08$.

**Desirability** ($D$) measures the effect of the observed action to help the user pursue the desirable goal safely. It separates common harmless actions from avoidable ones and connects the observations to knowledge of the goals the user wants to achieve. Given $\Pi_{candidates}$ as the set of plans extracted from the intervention graph that reach $G_d$ and $|\Pi_{candidates}| = n$. The plan set $\Pi_d$ contains action sequences that reach state $G_d$ without reaching $G_u$, $\Pi_d = \Pi_{candidates} \setminus \Pi_u$, we compute posterior probability of reaching $G_d$ without reaching $G_u$ for a path $\pi \in \Pi_d$, using chain rule in probability as, $P_\pi = \prod_{j=1}^{k} P(\alpha_j | \alpha_1, \alpha_2, ..., \alpha_{k-1})$, and $\alpha_j \in A$ and $k$ is the length of path. Then:

$$D = \begin{cases} \frac{\sum_{i=1}^{n-m} P_{\pi_i}}{n-m} & n - m > 0 \\ 0 & n - m = 0 \end{cases}$$

In Figure 5, there are five instances where user achieved

$G_d$ without reaching $G_u$ (two in subree $T_1$, three in the expanded branch). Extracting paths from root to these five instances, returns actions sequences the user may follow to reach $G_d$ safely ($\Pi_d$). Following the same approach to assign probabilities for states and actions, $D = \frac{(0.08+0.08+0.08+0.04+0.04)}{5} = 0.07$. Computation for $R$ and $D$ is similar for the passive attacker case.

$R$ and $D$ are based on probabilities indicating the confidence the observer has about the next observation. We also use simple distance measures: (1) distance to $G_u$ ($\delta_u$) and (2) distance to $G_d$ ($\delta_d$). Both distances are measured in the number of actions required to reach a state containing $G_d$ or $G_u$ from root in the intervention graph.

**Distance to $G_u$ ($\delta_u$)** measures the distance to state $G_u$ from the current state in terms of the number of actions. As with the computations of $R$ and $D$, given $\Pi_{candidates}$ is the set of paths extracted from the intervention graph that reach $G_d$ and $|\Pi_{candidates}| = n$. The path set $\Pi_u$ contains action sequences that reach state $G_u$ such that, $\Pi_u \subseteq \Pi_{candidates}$, $|\Pi_u| = m$ and $(m <= n)$. We count $s$, the number of the edges (actions) before $G_u$ is reached for each path $\pi \in \Pi_u$ and $\delta_u$ is defined as the average of the distance values given by the formula:

$$\delta_u = \begin{cases} \frac{\sum_{i=1}^{m} s_i}{m} & m > 0 \\ -1 & m = 0 \end{cases}$$

In this formula, $-1$ indicates that the undesirable state is not reachable from the current state. For the example problem illustrated in Figure 5, $\delta_u = \frac{3}{1} = 3$.

**Distance to $G_d$ ($\delta_d$)** measures the distance to $G_d$ from current state. The path set $\Pi_d$ contains action sequences that reach $G_d$ without reaching $G_u$, $\Pi_d = \Pi_{candidates} \setminus \Pi_u$, we count $t$, the number of the edges where $G_d$ is achieved without reaching $G_u$ for each path $\pi \in \Pi_d$. Then, $\delta_d$ is defined as the average of the distances given by the formula:

$$\delta_d = \begin{cases} \frac{\sum_{i=1}^{n-m} t_i}{n-m} & n - m > 0 \\ -1 & n - m = 0 \end{cases}$$

In this formula, $-1$ indicates that $G_d$ can not be reached safely from the current state. For the example problem illustrated in Figure 5, $\delta_d = \lceil \frac{3+3+7+7+3}{5} \rceil = 5$. Both $\delta_u$ and $\delta_d$ are computed similarly for the passive attacker case.

**Percentage of active attack landmarks ($\mathcal{L}_{ac}$)** captures the criticality of current state toward contributing to $G_u$. Landmarks (Hoffmann, Porteous, and Sebastia 2004) are predicates (or actions) that must be true in every valid plan for a planning problem. We used the algorithm in Hoffmann et al. (2004) to extract fact landmarks for the planning problem $P = \langle D, G_u \rangle$. These landmarks are referred to as attack landmarks because they establish predicates that must be true to reach $G_u$. Landmark Generation Graph (LGG) (Hoffmann, Porteous, and Sebastia 2004) for $P$ for the active attacker case is shown in Figure 6. Predicates (ON B A), (ON A D) correspond to $G_u$. Predicates that are grouped must be made true together. When the observed actions activate any attack landmarks, it signals that an undesirable state is imminent. Landmarks have been successfully used in deriving heuristics in plan recognition (Vered et al. 2018)

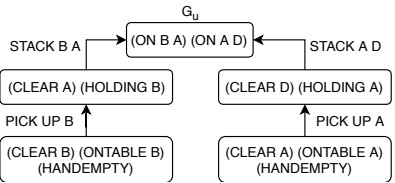

Figure 6: LGG for $P$. Contains verified fact landmarks for $P$ and *greedy-necessary orders*. A box with multiple landmarks indicate fact landmarks that must be true together.

---

**Algorithm 2** Generate Feature Vectors

---

**Require:** $D, I, O, G_u, G_d, p$-probability distribution.
1: **procedure** FEATUREVECTOR($D, O, I, G_u, G_d, p$)
2:     $i = 0; s_i \leftarrow I$
3:     **for** $o \in O$ **do**
4:         $G(V, E) \leftarrow ExpandGraph(D, s_i, G_u, G_d)$
5:         Apply action probabilities to $e \in E$ following $p$
6:         Apply state probabilities to $v \in V$ following $p$
7:         $V(o) \leftarrow [R_o, D_o, \delta_{u_o}, \delta_{d_o}, \mathcal{L}_{ac_o}, Class]$

---

and generating alternative plans (Bryce 2014). We compute a feature using attack landmarks: percentage of active attack landmarks in current state ($\mathcal{L}_{ac}$). To compute $\mathcal{L}_{ac}$ for the example in Figure 5, we count the number of landmark predicates that have become active ($l$) in the root of the intervention graph. Then, ($\mathcal{L}_{ac}$) is given by the formula: $\mathcal{L}_{ac} = \frac{l}{|\mathcal{L}_u|}$ In Figure 5, $l = 4$ ((CLEAR B),(CLEAR D),(ONTABLE B),(HOLDING A)) and $\mathcal{L}_{ac} = 4/10 = 0.4$.

## 4 Learning When to Intervene

We train the decision tree classifier in supervised learning mode to categorize observed actions into two classes: "Y" indicating that the interruption is warranted and "N", indicating that intervention is unwarranted. According to this policy, in the expanded sub-tree in Figure 5 the path that reaches $G_u$ is labeled as follows: PICK-UP A (N), STACK A D (N), PICK-UP B (N), STACK B A (Y). Label for each action is indicated within brackets. We will make this labeled data set available for the community. Given a labeled observation set and corresponding feature vectors, we train the decision tree classifier with 10-fold cross validation. Then the trained model is used to predict intervention for previously unseen intervention problems. We decided to chose the decision tree as the classifier because the decision tree learned model had the highest accuracy in predicting intervention on new problems compared to the two other classifiers: random forests (Breiman 2001) and Naive Bayes.

To generate training data we first created twenty planning problems for each benchmark domain. Then observation traces corresponding to each problem were generated. We enforced a limit of 100 observation traces for each planning problem for grid domains. These observation traces were provided as input to Algorithm 2. The algorithm takes a PDDL domain, a set of undesirable and desirable states and a probability distribution as input and produces a rela-

tion $V$ of observations and feature vectors. We train a decision tree classifier using the Weka [1] framework. We selected the implementation of C4.5 algorithm (Quinlan 1993) (J48), which builds a decision tree using the concept of information entropy. We chose the decision tree classifier for its ability determine salient features for intervention, which facilitates generating explanations for the user.

## 5    Results and Discussion

We focus on two questions: (1) Using domain-independent features indicative of the likelihood to reach $G_u$ from current state, can the intervening agent correctly interrupt to prevent the user from reaching $G_u$? and (2) If the user was not interrupted now, how can we establish a wait time until the intervention occurred before $G_u$? To address the first question, we evaluated the performance of the learned model to predict intervention on previously unseen problems.

The experiment suit consists of the two example domains from Section 2. To this we added Navigator and Ferry domains from IPC benchmarks. In Navigator domain, an agent simply moves from one point in grid to another goal destination. In the Ferry domain, a single ferry moves cars between different locations. To simulate intervention in active attacker case (the Block-Words domain), we chose word building problems. The words user and the attacker want to build are different but they have some common letters (e.g., TAD/BAD). The attacker is able to exploit the user's progress on stacking blocks to complete word the attacker wants to build. In Easy-IPC and Navigator domains, we designated certain locations on the grid as traps. The goal of the robot is to navigate to a specific point on the grid safely. In the Ferry domain a port is *compromised* and a ferry carrying a car there results in an undesirable state. The ferry's objective is to transport cars to specified locations without passing a compromised port.

In addition to the trained data set, we also generated 3 separate instances of 20 problems each (total of 60) for the benchmark domains to produce testing data for the learned model. The three instances contained intervention problems that were different the trained instances. For example, number of blocks in the domain (block-words), size of grid (navigator, easy-ipc), accessible and inaccessible paths on the grid (navigator, easy-ipc), properties of artifacts in the grid (easy-ipc). For each instance we generated 10 observation traces for each planning problem (i.e., 200 observation traces per instance). We define true-positive as the classifier correctly predicting "Y". True-negative is an instance where the classifier correctly predicts "N". False-positives are instances where classifier incorrectly predicts an observation as an interrupt. False-negatives are instances where the classifier incorrectly predicts the observation not as an interrupt.

### 5.1    Feature Selection

When a human user receives an interruption, the user may like to know a reason. To extract salient features for intervention, we applied a correlation based feature selection technique in data pre-processing step to identify the top four

---

[1] http://www.cs.waikato.ac.nz/ml/weka/

| Domain | Feature | Correlation |
|--------|---------|-------------|
| Blocks | Risk | 0.85 |
| | Distance to $G_d$ | 0.30 |
| | Desirability | 0.23 |
| | Distance to $G_u$ | 0.09 |
| Easy-IPC | Risk | 0.84 |
| | Distance to $G_d$ | 0.44 |
| | Distance to $G_u$ | 0.27 |
| | Desirability | 0.23 |
| Navigator | Risk | 0.85 |
| | Distance to $G_d$ | 0.28 |
| | Desirability | 0.18 |
| | Distance to $G_u$ | 0.04 |
| Ferry | Risk | 0.84 |
| | Distance to $G_d$ | 0.34 |
| | Desirability | 0.16 |
| | Distance to $G_u$ | 0.08 |

Table 1: Correlation factors of top 4 features for benchmark domains.

best predictors. Feature selection reduces complexity of the model, makes the outcome of the model easier to interpret, and reduces over-fitting.

The attribute selector in Weka uses the Pearson's correlation to measure predictive ability between nominal attributes and the class. Our feature vector consists of nominal attributes. Table 1 summarizes top 4 correlated features for each domain. Risk is the best performing feature. Distance desirable state feature is the next best choice for a feature. The percentage of active attack landmarks was the weakest predictor of intervention across all benchmark domains and was removed from training.

**Interrupting at each observation:** Assuming the decision to intervene is made for every observation, we calculated the true-positive rate (TPR=$\frac{TP}{TP+FN}$), false-positive rate (FPR=$\frac{FP}{FP+TN}$), true-negative rate (TNR= $\frac{TN}{TN+FP}$), false-negative rate (FNR=$\frac{FN}{TP+FN}$) of the trained model. For each domain, row 'Each' in table 2 summarizes TPR, FPR, TNR, FNR for predicting intervention in unseen observation traces. The classifier works well in identifying intervention across domains. In line with our expectation, TPR and TNR are very high ($> 95\%$) across domains and FNR and FPR is very low($< 5\%$). Because the accuracy remains consistant across test instances we conclude that the model is reasonably tolerant for modifications in the domain such as grid sizes and number of objects.

**Delaying the interruption:** In real-life, making the intervention decision for every observation may be costly. If we are intervening a human user, he may disregard frequent interruptions as noise. For this reason, we examine how to establish a wait time until intervention occurs for the first time. We used the feature ($\mathcal{L}_{ac}$) as a checkpoint for the intervening agent to wait safely without interrupting the user.

We modified the observation traces to contain action sequences starting from a point where the current state contained 50% and 75% of active landmarks. For problem instances where 75% active landmark percentage was infeasible, we limited it to the maximum active landmark percentage. We used the same learned model to predict intervention for these modified traces. For each domain, row 'Delayed50'

in table 2 summarizes TPR, FPR, TNR, FNR for predicting interruptions for benchmark domains given that the decision is delayed until $50\% <= \mathcal{L}_{ac} < 75\%$. The row 'Delayed75' indicates that the decision was delayed until $\mathcal{L}_{ac} >= 75\%$.

Accuracy is not affected significantly by delaying the intervention from the chosen checkpoints. However, a negative effect of delaying intervention is missing true positives. We evaluated how the delay affects the percentage of true positive observations missed. Table 3 summarizes these results. Intuitively, the longer the delay, a higher percentage of true positives will be missed. For the Blocks-Word domain, there is no effect between the the delay until 50% and 75%. In both cases the delaying the decision does not cause the intervening agent to miss any true positives. The most significant loss occurs in Navigator domain, where delay until 75% will cause a loss of 2%-28% while delaying until 50% is the safest choice. The Ferry domain exhibits a similar pattern where the delay until 75% landmarks become active will cause a loss of 8%-18%. We conclude that delaying interruptions can be controlled by the percentage of active landmarks in the current state and that for certain domains it is a trade off between loss of true-positives and the delay.

## 6 Explaining Intervention

When an observation that warrants intervention is identified intervening agent issues a warning (and an explanation) to the user. The user needs to take corrective/mitigating actions to avoid the undesirable state. The decision trees can help explain intervention. Decision trees generated for the benchmark domains are shown in Figure 7. Combining the shallow trees and the definitions of the features allow us to generate a clear and succinct set of rules to explain intervention. For the Block-word domain, (Figure 7-a), the rule that explains intervention first looks at the value of Risk. If the risk is less than or equal to 0.5 then that observation does not qualify as an intervention point. By definitions, this means that from the current state there are multiple ways to reach the undesirable state, indicating the observation is a common action that can be perceived as harmless. Next, if the observation that has a risk level of grater than 0.5 (indicating there are fewer ways of reach the undesirable state and that it's imminent), next feature to look at is the distance to the undesirable state. If the distance is negative, indicating that execution of this step will trigger the undesirable state, then the observation warrants intervention. Otherwise the observation does not require intervention. With this decision tree, an explanation for intervention in Blocks-words domain can be developed as: *The current step was intervened because the risk level is significant (> .5) and the effect of this observed action will trigger the undesirable state*.

For the passive attacker domains (Figure 7 - (b),(c),(d)) the learned model generated even simpler trees with only one feature being used to determine intervention. For Easy-IPC and Navigator domains, the Risk feature determines the class of an observation. This leads to generating explanations for the Easy-IPC and Navigator domains such as *The current step was intervened because the risk level is significant (> .75 for Easy-IPC and > .5 for Navigator)*. For the

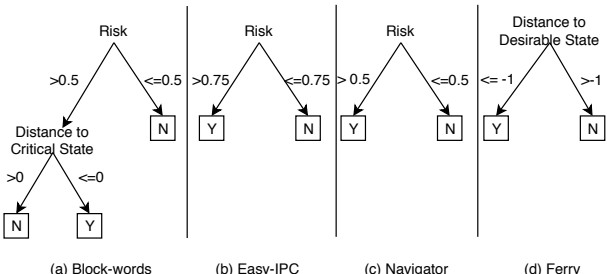

Figure 7: Decision trees generated for (a) Blocks, (b) Easy-IPC, (c) Navigator, (d) Ferry domains

Ferry domain, Distance to $G_d$ determines intervention. A negative value indicates that if the next step was executed there is no way to reach the desirable goal state without triggering the undesirable state. Thus an explanation of intervention for the Ferry domain will be: *The current step was intervened because the effect of this step will make it impossible to reach the desired goal without triggering the undesirable state*.

## 7 Related Work

Closely related areas of literature for this work is plan/goal recognition. Plan recognition is the problem of inferring the course of action (i.e., plan) an actor may take towards achieving a goal from a sequence of observations (Schmidt, Sridharan, and Goodson 1978; Kautz and Allen 1986). The constructed plan, if followed to completion, is expected to result in states that correspond to goals of the actor, which in turn presupposes that the actor intends to achieve those goals. Plan/goal recognition approaches in the literature explore both domain-dependent and independent methods. In domain-dependent methods agents rely heavliy on domain knowledge for inference. For example, Kabanza et al. (2010) presents a solution that recognizes an agents adversarial intent by mapping observations made to date to a plan library. Boddy et al. (2005) discuss how to construct and manipulate domain models that describe behaviors of adversaries in computer security domain and use these models to generate plans. Another approach uses Goal Driven Autonomy (GDA) that allows agents to continuously monitor the current plans execution and assess if the current state matches with expectation (Klenk, Molineaux, and Aha 2013).

More recent work attempts to separate this knowledge dependency by allowing the agent to learn knowledge from observations (Jaidee, Muñoz-Avila, and W. Aha 2011). In contrast, domain-independent goal recognition that use planning to infer agents goals. Ramirez and Geffner (2009; 2010) used an existing planner to generate hypotheses from observations to infer a single agent's plan. Their approaches offer advantages of being more adaptive to input as well as exploiting existing planning systems and plan representations. Their first approach computed the set of goals that can be achieved by optimal plans that match the observations. The second approach removed the optimality constraint and computed a probability distribution across possible plans

| Domain | Interrupt Type | Instance 1(20) | | | | Instance 2 (20) | | | | Instance 3 (20) | | | |
|---|---|---|---|---|---|---|---|---|---|---|---|---|---|
| | | TPR | FPR | TNR | FNR | TPR | FPR | TNR | FNR | TPR | FPR | TNR | FNR |
| Blocks | Each | 1 | 0 | 1 | 0 | 1 | 0 | 1 | 0 | 1 | 0 | 1 | 0 |
| | Delayed50 | 1 | 0 | 1 | 0 | 1 | 0 | 1 | 0 | 1 | 0 | 1 | 0 |
| | Delayed75 | 1 | 0 | 1 | 0 | 1 | 0 | 1 | 0 | 1 | 0 | 1 | 0 |
| Easy-IPC | Each | 1 | .05 | .95 | 0 | 1 | .03 | .97 | 0 | 1 | .03 | .97 | 0 |
| | Delayed50 | 1 | .06 | .94 | 0 | 1 | .03 | .97 | 0 | 1 | .03 | .97 | 0 |
| | Delayed75 | 1 | .06 | .94 | 0 | 1 | .03 | .97 | 0 | 1 | .03 | .97 | 0 |
| Navigator | Each | 1 | .01 | .99 | 0 | 1 | .03 | .97 | 0 | 1 | .02 | .98 | 0 |
| | Delayed50 | 1 | .01 | .99 | 0 | 1 | .03 | .97 | 0 | 1 | .02 | .98 | 0 |
| | Delayed75 | 1 | .02 | .98 | 0 | 1 | .03 | .97 | 0 | 1 | .03 | .97 | 0 |
| Ferry | Each | 1 | .02 | .98 | 0 | 1 | .05 | .95 | 0 | 1 | 0 | 1 | 0 |
| | Delayed50 | 1 | .02 | .98 | 0 | 1 | .05 | .95 | 0 | 1 | 0 | 1 | 0 |
| | Delayed75 | 1 | .02 | .98 | 0 | 1 | .03 | .97 | 0 | 1 | 0 | 1 | 0 |

Table 2: True-positive (TPR), False-positive (FPR), True-negative (TNR), False-negative (FNR) rates for predicting interrupt decision for unseen problems.

| Domain | Delay | Instance 1 | Instance 2 | Instance 3 |
|---|---|---|---|---|
| Blocks | Delayed50 | 0% | 0% | 0% |
| | Delayed75 | 0% | 0% | 0% |
| Easy-IPC | Delayed50 | 0% | 6% | 5% |
| | Delayed75 | 0% | 6% | 5% |
| Navigator | Delayed50 | 0% | 0% | 0% |
| | Delayed75 | 28% | 2% | 4% |
| Ferry | Delayed50 | 6% | 5% | 0% |
| | Delayed75 | 11% | 8% | 18% |

Table 3: Percentage of missed observations that should have been flagged as an interrupt

that could be generated from existing planners (Ramırez and Geffner 2010). Keren et al. (Keren, Gal, and Karpas 2014) introduced the worst-case distinctiveness (wcd) metric as a measurement of the ease of performing goal recognition in a domain. The wcd problem finds the longest sequence of actions an agent can execute while hiding its goal. They show that by limiting the set of available actions in the model wcd can be minimized, which will allow the agent to reveal it's goal as early as possible.

In online recognition, Vered et al. (2018) propose an approach that combines goal-mirroring and landmarks to infer the goal of an agent. Landmarks are used to minimize the number of hypotheses the agent has to evaluate, thus improving the effeciency of the recognition process. Pozanco et al. (2018) combines Ramirez and Geffener's plan recognition approach and leverages landmarks to counterplan and block an opponent's goal achievement. The main difference between plan intervention and recognition is that, in intervention the time intervention happens is critical. In plan recognition, identifying the plan at the right time is not a priority. The user's preferences in intervention (e.g., in-time, targetted intervention vs. prolonged and incremental) and the source of uncertainty in the environment (e.g., environment, attacker) complicate the intervening agent's decisioni and can be seen as trade-offs. Furthermore, our approach complements existing approaches by using a decision tree

to identify events that warrant intervention and identifying salient features that may be useful in generating explanations to plan intervention.

## 8 Summary and Future Work

We formalized the online plan intervention problem in a competitive domain where an attacker both actively and passively attempts to leverage progress made by a user to achieve the attacker's own conflicting goals. We introduced the intervention graph, which models the decision space of an observer, whose goal is to support the user by blocking actions that allows the attacker to achieve his goal. We trained a classifier using domain-independent features extracted from the intervention graph to evaluate the criticality of the current state. The model predicts intervention with high accuracy for the benchmark domains.

Our solution suffers from state space explosion for large domains. As an solution, we suggest sampling from alternative plans generated from off-the-shelf planners. This will also allow us to compare the proposed approach with existing online goal-recognition methods. The uncertainty model can be extended to limiting the observer's ability to fully perceive the current state. We recognize the attack models (for both active and passive cases) can be expanded to different threat models. For example, the attacker can behave as truly adversarial and undo progress the user has made so far and guide the user towards an entirely different goal. We will improve on explanations by suggesting actions that will help the user avoid the undesirable state when intervention occurs, instead of delegating the responsibility of being safe to the user, and integrating causal reasoning to explanations. These extensions lay a foundation for applying classical planning techniques for decision support and assistive agents.

## Acknowledgments

We thank the anonymous reviewers for comments that helped improve the paper. The authors also thank AFOSR and NRL for funding this research.

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
