# OpenReview forum: "Domain-independent Plan Intervention When Users Unwittingly Facilitate Attacks"
_icaps-conference.org/ICAPS/2019/Workshop/XAIP — XAIP 2019_

### Official Review · AnonReviewer2 · 2019-05-09
**Interesting paper but had a hard time situating it among the state of the art**

**Rating:** 3
**Confidence:** 3

**Review:**

The paper looks at the problem of computing interventions from an external observer in the plan of an actor in the presence of an active or passive adversary. I had some trouble situating the work (though the related works section at the end certainly helps).

> Assumptions: The assumptions made in the framework should be clearly stated. Section 3 starts with a list of them, but this is largely incomplete (e.g. some more appear in later sections and some more are implicit and are never mentioned at all). I would strongly suggest laying out what all the assumptions on each of the three agents are and how that impacts the intervention framework. Are the agents taking turns?

The actor seems to be a vanilla optimal agent, while the observer is bearing the computational burden of having to reason about attacks. This is different from agents that directly reason about the observer model (c.f. The landscape of Interpretable Agent Behavior [Chakraborti et al. ICAPS 2019]); This is the same trade-off in goal recognition design as well (with behaviors like legibility). I wonder what the differences are in the context of the intervention setting? Would be good to have that discussion in the related works.

Why are G_d and G_u part of the input? Given that the observer has the full model of both agents, interventions are exactly and precisely computable (c.f. Optimal Interdiction of Attack Plans [Letchford and Vorobeychik]). Why generate features and then classify? Is this purely a computational consideration or am I missing something?

Finally, and this is minor, it feels very strange to see "we use machine learning to determine..." in a technical paper. This is akin to "we used AI" in a magazine. Why not just say what it is? A classifier, a decision tree, etc.

---

### Official Review · AnonReviewer4 · 2019-05-13
**interesting problem, relation to XAI could be emphasized more**

**Rating:** 2
**Confidence:** 2

**Review:**


The paper introduces the problem of intervening, with an observer, to help an agent defend against an attacker. To my knowledge, this is a new problem formulation, and intuitively it makes sense.

The authors could do a better job at motivating their problem with appication scenarios. The illustrative examples follow IPC benchmarks, which is good to illustrate theor definitions, but which is not good to motivate why their problem is important. Can a use case be constructed in security testing, e.g. along the lines of Boddy et al? Or military applications? Logistics?

Regarding the XAIP workshop, the paper does contain a section on explaining the intervention to the agent being supported. This is great, but has a rather limited role in this paper. I'm not convinced it makes for a useful XAIP workshop to present whole host of papers each addressing completely different planning-related problems, each wth their idiosyncratic reference to explainability. If accepted, I urge the authors to expand on this aspect in their presentation at the workshop. From what I can gather from the paper, essentially their technique relies on the use of decision trees, and small ones at that. Presumably this is a standard technique? Is ther anything novel about this particular aspect of the work?

Overall I rate this paper to be at the borderline. Inclusion will depend on strength of the competition, and the amount of space/time available for presentations.

---

### Official Review · AnonReviewer1 · 2019-05-14
**Good Paper. Part on ML might be improved**

**Rating:** 3
**Confidence:** 3

**Review:**

The paper formalises the problem of attack prevention in a three-agents scenario (attacker, user and interventing/monitoring agent) in case of knowledge disparity between user and attacker that might facilitate the attack. The paper formalises the interventing/monitoring agent behaviour as a planning problem, using machine learning to lear how to intervene. The paper is well written, discussed, and formalised.
As a plus, the authors use domain-independent features (Risk, Desirability, Distance to D_g/D_u and Percentage on landmarks) that are formalised and that can be evaluated for each domain.

COMMENTS:

-- Given the nature of the paper, I was expecting to read more on the ML side, for example:
	-- Why a decision tree? Why you didn't apply classical ML algorithms to evaluate their performances?
	-- Authors evaluate TN/TP/FN/FP, but F1-score is a metric that considers all these variables and should be used to measure accuracy of a classification system
	-- did you use a grid-search to fine-tune parameters?
	-- did you use a training/test/validation phase?

-- On the explainability part, though DT are a common approach to understand how a black-box and opaque system works, they do not represent an explanation by themselves, as they do not encode the causality, that is a key element of explanations. I think the authors might plan to employ global interpretable models to explain the system (see, e.g., Guidotti, Riccardo, et al. "A survey of methods for explaining black box models." ACM computing surveys (CSUR) 51.5 (2018): 93.)

---

### Decision · Program_Chairs · 2019-05-15

**Decision:**

Accept

**Comment:**

While the reviewers view this paper somewhat critically, in the spirit of making the workshop a venue for discussion and feedback we decided to reject only those papers with strong reject votes.

Please address the review criticism as best possible for the final paper version and its presentation at the workshop. In particular, please carefully discuss the relation to/links to XAIP, and the XAIP literature. Looking forward to discuss your work at the workshop!